# Analysis of symbiotic backscatter empowered wireless sensors network with short-packet communications

**Quang Vinh Do[1], Bui Vu Minh[2], Quang-Sang Nguyen[3], Byung-seo Kim[4]***

1 Wireless Communications Research Group, Faculty of Electrical and Electronics Engineering, Ton Duc Thang University, Ho Chi Minh City, Vietnam, 2 Faculty of Engineering and Technology, Nguyen Tat Thanh University, Ho Chi Minh City, Vietnam, 3 Posts and Telecommunications Institute of Technology, Ho Chi Minh City, Vietnam, 4 Department of Software and Communications Engineering, Hongik University, Sejong, South Korea

* jsnbs@hongik.ac.kr

**Data Availability Statement:** We confirm that "All relevant data are within the manuscript and its Supporting information files." We have also

## Abstract

Recent progress studies in light of wireless communication systems mainly centred around two focuses: zero-energy consumption and ultra-reliable and low-latency communication (URLLC). Among various cutting-edge areas, exploiting ambient backscatter communication (Backcom) has recently been devised as one of the foremost solutions for achieving zero energy consumption through the viability of ambient radio frequency. Meanwhile, using short-packet communication (SPC) is the cheapest way to reach the goal of URLLCs. Upon these benefits, we investigate the feasibility of Backcom and SPC for symbiotic wireless sensor networks by analyzing the system performance. Specifically, we provide a highly approximated mathematical framework for evaluating the block-error rate (BLER) performance, followed by some useful asymptotic results. These results provide insights into the level of diversity and coding gain, as well as how packet design impacts BLER performance. Numerical results confirm the efficacy of the developed framework and the correctness of key insights gleaned from the asymptotic analyses.

## 1 Introduction

With the widespread adoption of the Internet-of-Things (IoT), next-generation wireless networks are seeing an influx of emerging IoT applications and services [1]. These applications and services are not limited to interactions between a specific system, such as wireless sensor networks (WSNs), cellular networks, or vehicle networks, but also extend to new categories such as telemetry, healthcare, smart grids, digital twins, and the metaverse [2]. In which, artificially intelligent technology plays a core role in controlling, collaborating and coordinating network components [3]. Along with the benefits brought by IoT context, it also renders several challenges to the development of physical wireless systems [4]. For instance, how to accommodate the increasing demand for connected devices when the current spectrum is full,

provided our simulation code as Supporting information Files.

**Funding:** This work was supported by the National Research Foundation of Korea (NRF) grant funded by the Korea government (MSIT) (No. 2022R1A2C1003549).

**Competing interests:** The authors have declared that no competing interests exist.

prolong the activities of IoT devices with limited power resources, and provide ultra-reliable and low-latency wireless communication (URLLC).

Amidst the shortage of available radio frequencies, several efficient strategies have been proposed to tackle this challenge. The first aim is to line on full-duplex communication to increase spectrum efficiency [5] but requires refined successive interference cancellation (SIC) approaches to move residual interference it produces [6, 7]. Another strategy is to take advantage of multiple access technologies, such as non-orthogonal multiple access [8] and rate-splitting multiple access [9], which enable multiple users to communicate simultaneously using the same time and frequency resource block. However, exploiting such multiple access is tedious because it is only suitable for a single network. As a result, recent progress on multiple access typically combines it with two potential types of cognitive radio (CR) paradigms [10, 11], overlay and underlay. While underlay allows the secondary network to coexist with the primary network subject to power-tolerant constraints, overlay networks operate under free interference [12]. However, with the disadvantage characteristics by themselves, the realization of CR solutions also becomes controvertible [13, 14].

In the wake of energy constraints, energy harvesting (EH) solutions have emerged as an alternative that allows IoT devices to charge their batteries without human intervention [15]. Compared to charging energy from surrounding natural sources such as wind or sun, radio-frequency (RF) EH is voted as the most cost-effective solution, with two representations of wireless-power transfer communication network and simultaneously wireless information power transfer [16]. According to many reports in the literature [17–22], the nature of active communication enabled by RF signals consumes relatively high power, and this might not be favourable to large-scale IoT deployments in the long term. Thus, it raises the question of finding new alternatives with sustainable RF-EH capabilities and low power consumption.

Driven by the two necessities above, symbiotic communication, a new paradigm shift for passive IoT, has recently emerged as the ultimate solution to tackle the issues of spectrum scarcity and low-power consumption [23]. Symbiotic paradigm is a revolutionary concept that expertly blends the strengths of two existing paradigms [24], ambient backscatter communication (Backcom) and CR. In this paradigm, the backscatter device passively modifies the received signal from the primary transmitter with its information before sending it back to the secondary receiver [25], effectively changing its load impedance instead of using dedicated RF components. This allows Backcom to act similarly to CR paradigms while consuming zero energy. This functionality helps a symbiotic paradigm to achieve properties of mutualism, commensalism or parasitism [26]. Due to this prominent feature, the research on Backcom with symbiotic mediums has recently attracted significant momentum from both industrial and academia. For example, a full-duplex Backcom solution was proposed to symbiotic radio (SR) system [27]. In [28], three practical cooperative transmission schemes was proposed for symbiotic radio systems. The work in [29] provided a thorough and authoritative review of the systematic view for SR, along with critical discussions to enhance the backscattering link, achieve highly reliable communications, and effectively utilize reconfigurable intelligent surfaces. In [30], a novel beamforming design was proposed to multiple-input-multiple-output SR backscatter system. Meanwhile, the work in [31] studied SR communication system in the presence of multiuser multi- backscatter-device. In [32], two SR schemes were designed for a pair of backscatter devices, is that, opportunistic commensal and opportunistic parasitic. In [33], a symbiotic localization and Backcom architecture was developed for IoT localize target objects to achieve two mutual benefits: sensing and communication stage. In [34], an investigation of Backcom was put forward in symbiotic cell-free massive multiple-input multiple-output systems. Meanwhile, an innovative solutions for enhancing the security of low-power IoT devices using ambient backscatter communication was introduced in [35], with a strong focus

on the balance between energy efficiency and security. In a very recent time, the work in [36] presented advances in enhancing the robustness of wireless communication systems against jamming attacks by designing a novel beamforming technique that utilizes the concept of symbiotic radio to effectively use the null space of interference, thereby enhancing safeguarding data transmission significantly.

On the other hand, to deal with stringent URLLC conditions, where transmission latency is expected to be less than 1 ms while reliability is from 99.9% to 99.9999%, recent efforts propose to rethink the design of packet size [37]. Specifically, reducing packet size to improve the physical layer transmission latency; however, this action results in a higher error rate transmission. In this case, there is no way to use a finite blocklength message coding scheme to boost reliable communication. Based on Polyanskiy's novel infinite block length theory, published in 2010 [38], the research on short-packet communication (SPC) has recently emerged as a vital solution and is receiving considerable interest from research communities [39–42]. In that, block-error rate (BLER) is devised to be the key metric instead of using outage probability or ergodic Shannon capacity for the performance evaluation.

Towards a green IoT network for the future, the interplay between symbiotic Backcom and SPC becomes the pivotal direction. In the past, several works investigated the benefits brought by SPC with conventional Backcom systems (backscatter devices are deployed for enhancing communication coverage only), such as resource allocation [43], energy efficiency [44], and error performance for finite backscatter channels [45]. However, to the best of authors' knowledge, the research on symbiotic Backcom systems with SPC remains unexplored in the literature. Therefore, this inspires us to investigate the feasibility of SPC in symbiotic Backcom systems. In particular, the main focus of this work is on the performance of symbiotic Backcom-empowered WSNs with SPC, where the secondary backscatter transmitter is parasitic from the primary network. Overall, the main contribution of this article can be outlined as follows:

- Towards future green URLLC use cases, this article studies the performance of symbiotic backscatter-empowered WSNs, where a passive backscatter device with energy constraints in the secondary networks exploits ambient RF signals generated by the primary transmitter for the primary receiver as a green power source to be able to communicate with the secondary IoT receiver. To reject interference impacted by a primary transmitter's RF signals, SIC enables the IoT receiver to decode its signal from a passive backscatter device.

- Aiming at characterizing the performance of the considered networks, we provide an efficiently approximated mathematical framework for the BLER performance evaluation without any simulations or empirical, where we first endeavour to seek a way to derive closed-form solutions for the signal-to-noise ratio (SNR) distributions received by the primary and secondary receivers, while putting all the remaining energies to the work of finding out the BLER approximation. Not only these, but we also provide some insightful asymptotic analyses, which allow us effortlessly to answer these critical questions:

  1. How much diversity and coding gains does the considered system achieve when compared to a system using uncoded transmission?

  2. How do the packet designs affect the BLER performance?

- To validate our developed mathematical framework, we provide some extensive numerical results based on Monte-Carlo simulations method. It is interesting to show that this framework accurately predicts the actual result with very minor errors, even with a series of approximation approaches used. Besides, it also confirms the performance trend findings of

the reflection coefficient designed at the backscatter device, the packet construction involving packet length and number of information bits, as well as the transmit power of the primary transmitter. Furthermore, we through numerical results have that when boosting the reflection coefficient exceeds 2.5 (unit), the BLER performance of the secondary IoT receiver converges to saturation.

The remainder of this article is covered as follows. Section 2 describes the system model, followed by the average BLER analysis in Section 3. Next, Section 5 provides some numerical results before concluding the article in Section 5.

## 2 System model

Let us consider a symbiotic backscatter communication system as shown in Fig 1, where the cellular network, called the primary network, coexists with an IoT sensor network, called the secondary network. In this setup, a backscatter device (named by BD) exploits the available RF signal when carrying a symbol $x(t)$ sent from the primary transmitter (denoted by PT) to the primary receiver (called PR) to convey its symbol information $c(t)$ to the secondary IoT sensor receiver (i.e., IR), with $t$ being the time $t$. In which, the packet information sent by BD has data amount $N_{IR}$ bits with packet length $L_{IR}$ (or the equivalent terminologies: channel use or block-length), while that of PT includes data amount $N_{PR}$ bits with packet length $L_{PR}$. Due to the presence of the multiplicative fading phenomenon and long-distance communication, no interference occurs from BD to the signal of PR [27, 29, 33]. Meanwhile, there always exists interference from PT to IR, which therefore requires the adoption of SIC approach at IR to subtract $x$ from the received signal before detecting $c(t)$. In this investigation, all wireless channels are assumed to follow quasi-static Rayleigh block fadings, which means that channels vary very small or even with static. Thus, it is reasonable to consider the availability of global channel state information at the terminals via statistical channel measurement methods [39–42, 45].

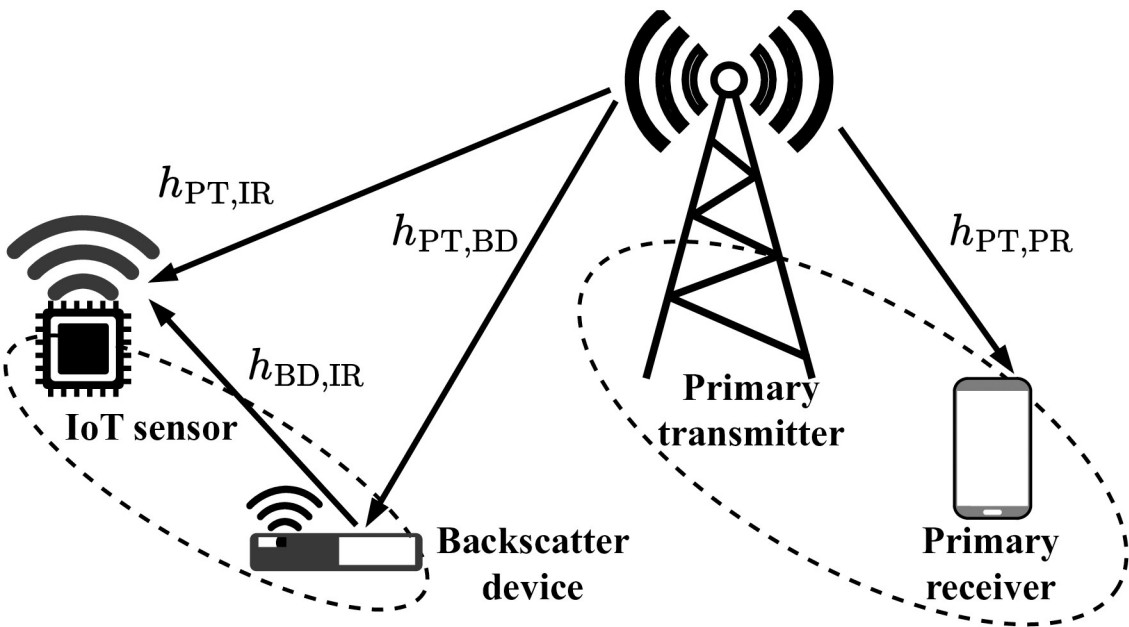

**Fig 1. Illustration of the considered system.**

For the sake of analysis, we denote by $h_{PT,PR}$, $h_{PT,BD}$, $h_{PT,IR}$, and $h_{BD,IR}$ the channels of links $PT \rightarrow PR$, $PT \rightarrow BD$, $PT \rightarrow IR$, and $BD \rightarrow IR$, respectively. Under Rayleigh fading assumptions, channel gains $|h|^2$, $h \in \{h_{PT,PR}, h_{PT,BD}, h_{PT,IR}, h_{BD,IR}\}$ will obey exponential distributions with the parameter $\lambda \in \{\lambda_{PT,PR}, \lambda_{PT,BD}, \lambda_{PT,IR}, \lambda_{BD,IR}\}$, and the respective probability density function (PDF) and the cumulative distribution function (CDF) are given by [46]

$$f_{|h|^2}(x) = \lambda \exp(-\lambda x), \quad F_{|h|^2}(x) = 1 - \exp(-\lambda x). \tag{1}$$

Under the path-loss measurement, we can model $\lambda$ as $\lambda = d^\beta$ [47], with $d \in \{d_{PT,PR}, d_{PT,BD}, d_{PT,IR}, d_{BD,IR}\}$ is the physical distance of links $h$ and $\beta \in [2, 6]$ is the path-loss exponent.

Considering a block duration $T > 0$, the signal received at PR when sending $x$ from PT with the transmit power $P_{PT}$ over channel $h_{PT,PR}$ at the time $t \in T$ can be written as

$$y_{PR} = \sqrt{P_{PT}} h_{PT,PR} x(t) + n_{PR}(t), \tag{2}$$

where $n_{PR}$ is the additive White Gaussian noise (AWGN) with zero-mean and variance $N_0$ and $E\{|x(t)|^2\} = 1$, with $E\{\cdot\}$ being the expectation operator. On the foundation of (2), the signal-to-noise (SNR) ratio to decode $x(t)$ can be written as

$$\gamma_{PR} = \frac{P_{PT}}{N_0}|h_{PT,PR}|^2 = \Psi|h_{PT,PR}|^2, \quad \Psi \triangleq \frac{P_{PT}}{N_0}. \tag{3}$$

During the duration $T$, the communication between BD and IR goes through two stages. In particular, BD firstly collects the power signal from PT based on energy harvesting technology. Then, BD modulates its received signal power with $c(t)$ and then backscatters it back to IR. As a result, the signal received at IR at the time $t$ can be expressed as

$$\begin{aligned} y_{IR} &= \sqrt{P_{PT}} h_{PT,IR} x(t) + \sqrt{\alpha} h_{BD,IR} \sqrt{P_{PT}} h_{PT,BD} x(t - \tau) c(t) \\ &\quad + n_{IR}(t), \end{aligned} \tag{4}$$

where $\alpha \in (0, 1)$ is a reflection coefficient used to normalize $c(t)$, $\tau$ is the processing delay at BD, and $n_{IR}$ is the AWGN with zero-mean and variance $N_0$. In order to reduce the impact of interference, SIC is considered at IR with the decoding order $x(t) \rightarrow c(t)$. Thus, the corresponding signal-to-interference-plus-noise ratio (SINR) of decoding $x(t)$ and SNR of decoding $c(t)$ at IR can be written as

$$\gamma_{IR,x(t)} = \frac{\Psi|h_{PT,IR}|^2}{\alpha\Psi|h_{PT,BD}|^2|h_{BD,IR}|^2 + 1}, \tag{5}$$

$$\gamma_{IR,c(t)} = \alpha\Psi|h_{PT,BD}|^2|h_{BD,IR}|^2. \tag{6}$$

From the formula above, the signal at IR is said to be successfully decoded when $x(t)$ and $c(t)$ are perfectly decoded. Thus, the end-to-end SNR and SINR at IR can be claimed by

$$\gamma_{IR} = \min\{\gamma_{IR,x(t)}, \gamma_{IR,c(t)}\}. \tag{7}$$

## 3 Average BLER analysis

When a packet having blocklength $L$ is sent from the transmitter node to the receiver node with the SNR $\gamma$ and error probability $\epsilon$, Polyanskiy's novel infinite block length theory said that

the maximal channel coding ratio can be approximated as [38]

$$r \approx C(\gamma) - \sqrt{\frac{V(\gamma)}{L}} Q^{-1}(\epsilon),$$

(8)

where $C(x) = \log_2(1 + x)$ is the Shannon capacity, $V(x) = (\log_2(e))^2[1 - 1/(1 + x)^2]$ is the channel dispersion, and $Q^{-1}(x)$ is the inverse of the Gaussian Q-function, i.e., $Q(x) = \int_x^\infty \frac{1}{\sqrt{2\pi}} \exp(-z^2/z)dz$.

Provided that a packet has $N_{\text{Rx}}$ bits and packet length $L_{\text{Rx}}$, with $\text{Rx} \in \{\text{PR}, \text{IR}\}$ and invoking the relation in (8), the average BLER at Rx can be written as [39–42]

$$\epsilon_{\text{Rx}} = \int_0^\infty Q\left(\frac{C(x) - r_{\text{Rx}}}{\sqrt{V(x)/L_{\text{Rx}}}}\right) f_{\gamma_{\text{Rx}}}(x)dx,$$

(9)

with $r_{\text{Rx}} = N_{\text{Rx}}/L_{\text{Rx}}$. Unfortunately, the complex nature of the Q-function makes the mathematical analysis quite challenging. To end this, making use of the normal approximation method in [11] shows that $Q\left(\frac{C(x) - r_{\text{Rx}}}{\sqrt{V(x)/L_{\text{Rx}}}}\right)$ can be linearly approximated by $\Xi(r_{\text{Rx}}, L_{\text{Rx}}, \gamma_{\text{Rx}})$, i.e.,

$$\Xi(r_{\text{Rx}}, L_{\text{Rx}}, \gamma_{\text{Rx}}) = \begin{cases} 1, & \gamma_{\text{Rx}} \leq \upsilon_{\text{Rx}} \\ 0, & \gamma_{\text{Rx}} \geq \vartheta_{\text{Rx}} \\ \frac{1}{2} - \zeta_{\text{Rx}}(\gamma_{\text{Rx}} - \kappa_{\text{Rx}}), & \text{otherwise}, \end{cases}$$

(10)

where $\zeta_{\text{Rx}} = \sqrt{L_{\text{Rx}}/[2\pi(2^{2r_{\text{Rx}}} - 1)]}$, $\kappa_{\text{Rx}} = 2^{r_{\text{Rx}}} - 1$, $\upsilon_{\text{Rx}} = \kappa_{\text{Rx}} - 1/(2\zeta_{\text{Rx}})$, and $\vartheta_{\text{Rx}} = \kappa_{\text{Rx}} + 1/(2\zeta_{\text{Rx}})$.

Pulling (9) and (10) together and applying the integral-by-parts method, the average BLER at Rx can be rearranged as

$$\epsilon_{\text{Rx}} \simeq \int_0^\infty \Xi(r_{\text{Rx}}, L_{\text{Rx}}, \gamma_{\text{Rx}}) f_{\gamma_{\text{Rx}}}(x)dx = \zeta_{\text{Rx}} \int_{\upsilon_{\text{Rx}}}^{\vartheta_{\text{Rx}}} F_{\gamma_{\text{Rx}}}(x)dx.$$

(11)

It is intuitively observed from (11) that finding the average BLER goes through two steps: 1) finding out the CDF of $\gamma_{\text{Rx}}$ and 2) deriving the average BLER based on the relation in (11). Consequently, we should recognize that most technical challenges come from finding the CDF characteristic, solving the integral of the average BLER or even both of them.

## 3.1 Average BLER analysis of Primary Receiver (PR)

**3.1.1 Statistical analysis of SNR distribution.** Based on the developed formula in (11), we in this subsection will focus on deriving the CDF of $\gamma_{\text{PR}}$, denoted by $F_{\gamma_{\text{PR}}}(x)$. Specifically, revising the SNR built-in (3), we can mathematically derive a closed-form solution for $F_{\gamma_{\text{PR}}}(x)$ as follows:

$$\begin{aligned} F_{\gamma_{\text{PR}}}(x) &= \Pr[\gamma_{\text{PR}} < x] = \Pr\left[|h_{\text{PT,PR}}|^2 < \frac{x}{\Psi}\right] \\ &= \int_0^{\frac{x}{\Psi}} f_{\gamma_{\text{PR}}}(x)dx = \int_0^{\frac{x}{\Psi}} \lambda_{\text{PR}} \exp(-\lambda_{\text{PR}}x)dx = 1 - \exp\left(-\frac{\lambda_{\text{PR}}}{\Psi}x\right). \end{aligned}$$

(12)

**3.1.2 Average BLER analysis.** Having obtained the CDF of $\gamma_{\mathrm{PR}}$ in hand, we are next so excited to derive the average BLER of PR by making use of the relation in (11).

**Theorem 1**. *A closed-form solution for the average BLER of the primary receiver is formulated as*

$$\epsilon_{\mathrm{PR}} \simeq 1 - \frac{\zeta_{\mathrm{PR}}\Psi}{\lambda_{\mathrm{PR}}} \left[ \exp\left(-\frac{\lambda_{\mathrm{PR}}}{\Psi}\vartheta_{\mathrm{PR}}\right) - \exp\left(-\frac{\lambda_{\mathrm{PR}}}{\Psi}\upsilon_{\mathrm{PR}}\right) \right]. \tag{13}$$

*Proof.* By injecting the developed CDF in (12) into the expression in (11), we can establish the following average BLER formulation

$$
\begin{aligned}
\epsilon_{\mathrm{PR}} &\simeq \zeta_{\mathrm{PR}} \int_{\upsilon_{\mathrm{PR}}}^{\vartheta_{\mathrm{PR}}} F_{\gamma_{\mathrm{PR}}}(x)dx = \zeta_{\mathrm{PR}} \int_{\upsilon_{\mathrm{PR}}}^{\vartheta_{\mathrm{PR}}} \left[ 1 - \exp\left(-\frac{\lambda_{\mathrm{PR}}}{\Psi}x\right) \right] dx \\
&= \zeta_{\mathrm{PR}}(\vartheta_{\mathrm{PR}} - \upsilon_{\mathrm{PR}}) - \zeta_{\mathrm{PR}} \int_{\upsilon_{\mathrm{PR}}}^{\vartheta_{\mathrm{PR}}} \exp\left(-\frac{\lambda_{\mathrm{PR}}}{\Psi}x\right) dx.
\end{aligned}
\tag{14}
$$

Applying the fact that $\vartheta_{\mathrm{PR}} - \upsilon_{\mathrm{PR}} = 1/\zeta_{\mathrm{PR}}$ and $\int \exp\left(-\frac{\lambda_{\mathrm{PR}}}{\Psi}x\right) = \frac{\Psi}{\lambda_{\mathrm{PR}}} \exp\left(-\frac{\lambda_{\mathrm{PR}}}{\Psi}x\right)$ for (14), we can obtain the final solution in (13). The proof is completed.

Having achieved Theorem 1, we are interested in concluding that the average BLER of PR can be characterized by a unique function involving all elementary functions. Thus, it is feasible to use common integrated software packages (i.e., Matlab, Maple, or Mathematica) to dissect the average BLER performance by this function without any simulation or actual testing. Yet, it would be extremely meaningful to explore or answer the questions of whether is there any simpler way to characterize the average BLER performance limits at a high SNR regime and how much performance gain the system can be achieved compared to an uncoded transmission.

To answer these questions, let us turn to evaluate the asymptotic BLER. To begin with, let us revisit the CDF of $\gamma_{\mathrm{PR}}$ in (12). By using the fact that $1 - \exp(-x) \simeq x$ as $x$ goes to 0, we can simplify the expression in (12) as

$$F_{\gamma_{\mathrm{PR}}}(x) \overset{\Psi \to \infty}{\simeq} \frac{\lambda_{\mathrm{PR}}}{\Psi}x. \tag{15}$$

Combining this approximation with the formulation in (11), the asymptotic BLER can be computed as

$$
\begin{aligned}
\tilde{\epsilon}_{\mathrm{PR}} &\overset{\Psi \to \infty}{\simeq} \zeta_{\mathrm{PR}} \int_{\upsilon_{\mathrm{PR}}}^{\vartheta_{\mathrm{PR}}} F_{\gamma_{\mathrm{PR}}}(x)dx = \zeta_{\mathrm{PR}} \int_{\upsilon_{\mathrm{PR}}}^{\vartheta_{\mathrm{PR}}} \frac{\lambda_{\mathrm{PR}}}{\Psi} x dx \\
&\overset{(*)}{=} \zeta_{\mathrm{PR}} \frac{\lambda_{\mathrm{PR}}}{2\Psi} \left[ \vartheta_{\mathrm{PR}}^2 - \upsilon_{\mathrm{PR}}^2 \right] = \lambda_{\mathrm{PR}} \frac{2^{r_{\mathrm{PR}}} - 1}{\Psi}.
\end{aligned}
\tag{16}
$$

Herein, the last step is obtained based on a basic equality that $x^2 - y^2 = (x + y)(x - y)$. Accordingly, we have that $\vartheta_{\mathrm{PR}}^2 - \upsilon_{\mathrm{PR}}^2 = (\vartheta_{\mathrm{PR}} - \upsilon_{\mathrm{PR}})(\vartheta_{\mathrm{PR}} + \upsilon_{\mathrm{PR}})$. Recall that $\upsilon_{\mathrm{Rx}} = \kappa_{\mathrm{Rx}} - 1/(2\zeta_{\mathrm{Rx}})$ and $\vartheta_{\mathrm{Rx}} = \kappa_{\mathrm{Rx}} + 1/(2\zeta_{\mathrm{Rx}})$, with $\zeta_{\mathrm{Rx}} = \sqrt{L_{\mathrm{Rx}}/[2\pi(2^{2r_{\mathrm{Rx}}} - 1)]}$ and $\kappa_{\mathrm{Rx}} = 2^{r_{\mathrm{Rx}}} - 1$, as provided in Eq (10). On that basis, we can obtain $\vartheta_{\mathrm{PR}} + \upsilon_{\mathrm{PR}} = 2\kappa_{\mathrm{PR}} = 2(2^{r_{\mathrm{PR}}} - 1)$ and $\vartheta_{\mathrm{PR}} - \upsilon_{\mathrm{PR}} = 1/\zeta_{\mathrm{PR}}$. Pulling all together yields $\vartheta_{\mathrm{PR}}^2 - \upsilon_{\mathrm{PR}}^2 = 2(2^{r_{\mathrm{PR}}} - 1)/\zeta_{\mathrm{PR}}$. By comparing this result with step $(*)$ in Eq (16), we can readily obtain the desired result.

From the formulation in (16), it is wonderful to show that the BLER performance of PR is dominated by three factors: the average SNR $\Psi$, the fading parameter $\lambda_{\mathrm{PR}}$, and the coding ration $r_{\mathrm{PR}}$. For the characteristic of $\Psi$, it is found that the BLER performance is proportional to $\Psi$, which concludes the diversity gain of 1 and the respective coding gain is $[\lambda_{\mathrm{PR}}(2^{r_{\mathrm{PR}}} - 1)]^{-1}$. For the characteristic of $\lambda_{\mathrm{PR}}$, we notice that when the distance parameter $d_{\mathrm{PT,PR}}$ increases, or PR moves far from the PT in other words, the ABLER increases, which completely accords the fact that the larger the communication coverage, the lower the performance quality. For the characteristic of $r_{\mathrm{PR}} = N_{\mathrm{PR}}/L_{\mathrm{PR}}$, we can readily observe from the definition that $r_{\mathrm{PR}}$ is an increasing function of $N_{\mathrm{PR}}$ but a decreasing function of $L_{\mathrm{PR}}$. This means that increasing the data amount makes the transmission more error. In contrast, increasing the block-length decreases the error data transmission, which then significantly enhances a reliable communication system.

### 3.2 Average BLER analysis of IoT sensor receiver

**3.2.1 Statistical analysis of SNR distribution.** Likewise, evaluating the average BLER at IR also requires the CDF of $\gamma_{\mathrm{IR}}$, denoted by $F_{\gamma_{\mathrm{IR}}}(x)$. Invoking the SNR built-in (7) and denoting by $Y \triangleq |h_{\mathrm{PT,BD}}|^2 |h_{\mathrm{BD,IR}}|^2$, we can express $F_{\gamma_{\mathrm{IR}}}(x)$ using the complementary probability property as follows:

$$
\begin{aligned}
F_{\gamma_{\mathrm{IR}}}(x) \;\; &= 1 - \Pr[\gamma_{\mathrm{IR}} \geq x] = \Pr[\min\{\gamma_{\mathrm{IR},x(t)}, \gamma_{\mathrm{IR},c(t)}\} \geq x] \\
&= 1 - \Pr\left[\frac{\Psi|h_{\mathrm{PT,IR}}|^2}{\alpha\Psi Y + 1} \geq x, \alpha\Psi Y \geq x\right] = 1 - \Pr\left[|h_{\mathrm{PT,IR}}|^2 \geq x\frac{\alpha\Psi Y + 1}{\Psi}, Y \geq \frac{x}{\alpha\Psi}\right] \\
&= 1 - \int_{\frac{x}{\alpha\Psi}}^{\infty} \Pr\left[|h_{\mathrm{PT,IR}}|^2 \geq x\frac{\alpha\Psi y + 1}{\Psi}\right] f_Y(y)dy \\
&= 1 - \int_{\frac{x}{\alpha\Psi}}^{\infty} \exp\left(-\frac{\alpha\Psi y + 1}{\Psi/\lambda_{\mathrm{PT,IR}}}x\right) f_Y(y)dy,
\end{aligned}
\tag{17}
$$

where the last step is achieved based on the relation $\Pr[|h_{\mathrm{PT,IR}}|^2 > x] = 1 - F_{|h_{\mathrm{PT,IR}}|^2}(x)$. Observing the above integral shows that to achieve the solution, the foremost important task is now to derive the joint PDF of $|h_{\mathrm{PT,BD}}|^2$ and $|h_{\mathrm{BD,IR}}|^2$. To proceed, let us consider the following derivation

$$
f_Y(y) = \frac{\partial F_Y(y)}{\partial y},
\tag{18}
$$

where the joint CDF of $Y$ can be obtained as [18]

$$
F_Y(y) = 1 - 2\sqrt{\lambda_{\mathrm{PT}\to\mathrm{IR}}y}\,\mathcal{K}_1(2\sqrt{\lambda_{\mathrm{PT}\to\mathrm{IR}}y}),
\tag{19}
$$

with $\lambda_{\mathrm{PT}\to\mathrm{IR}} \triangleq \lambda_{\mathrm{PT,BD}}\lambda_{\mathrm{BD,IR}}$ and $\mathcal{K}_1(\cdot)$ being the first order modified Bessel function of the second kind.

Next, plugging (19) into (18) combined with the relation $\frac{d}{dz}[z^m\mathcal{K}_m(z)] = -z^m\mathcal{K}_{m-1}(z)$ [48], we have that

$$
f_Y(y) = 2\lambda_{\mathrm{PT}\to\mathrm{IR}}\mathcal{K}_0(2\sqrt{\lambda_{\mathrm{PT}\to\mathrm{IR}}y}).
\tag{20}
$$

Having developed the PDF of $Y$ in hand, we can rewrite the CDF of $\gamma_{\text{IR}}$ by injecting (20) into (17), which yields

$$
\begin{aligned}
F_{\gamma_{\text{IR}}}(x) \quad &= 1 - 2\lambda_{\text{PT},\text{IR}} \exp\left(-\frac{\lambda_{\text{PT},\text{IR}} x}{\Psi}\right) \int_{\frac{x}{\alpha\Psi}}^{\infty} \exp(-\alpha\lambda_{\text{PT}\to\text{IR}} y x) \mathcal{K}_0(2\sqrt{\lambda_{\text{PT}\to\text{IR}} y}) dy \\
&= 1 - 2\lambda_{\text{PT},\text{IR}} \exp\left(-\frac{\lambda_{\text{PT},\text{IR}} x}{\Psi}\right) \int_{0}^{\infty} \exp(-\alpha\lambda_{\text{PT},\text{IR}} y x) \\
&\quad \times \mathcal{K}_0(2\sqrt{\lambda_{\text{PT}\to\text{IR}} y}) \mathcal{H}\left(\frac{\alpha\Psi y}{x} - 1\right) dy,
\end{aligned}
\tag{21}
$$

where $\mathcal{H}(\cdot)$ is the unit step function with

$$
\mathcal{H}(z) = \begin{cases} 1, z > 0, \\ 0, z < 1. \end{cases}
\tag{22}
$$

To get the final solution for (21), we conjure the three following transformations into the Meijer-G function as [49]

$$
\exp(-z) = G_{0,1}^{0,1}\left(z \,\middle|\, \begin{matrix} - \\ 0 \end{matrix}\right), \mathcal{H}(|z| - 1) = G_{0,1}^{1,1}\left(z \,\middle|\, \begin{matrix} 1 \\ 0 \end{matrix}\right),
\tag{23}
$$

$$
z^m \mathcal{K}_n(z) = 2^{m-1} G_{0,2}^{2,0}\left(\frac{1}{4}z^2 \,\middle|\, \begin{matrix} - \\ \frac{1}{4}m + \frac{1}{4}n, \frac{1}{4}m - \frac{1}{4}n \end{matrix}\right).
\tag{24}
$$

Applying the transformation into (21), we obtain

$$
\begin{aligned}
F_{\gamma_{\text{IR}}}(x) &= 1 - \lambda_{\text{PT}\to\text{IR}} \exp\left(-\frac{\lambda_{\text{PT},\text{IR}} x}{\Psi}\right) \int_{0}^{\infty} G_{1,1}^{0,1}\left(\frac{\alpha\Psi y}{x} \,\middle|\, \begin{matrix} 1 \\ 0 \end{matrix}\right) \\
&\quad \times G_{0,2}^{2,0}\left(\lambda_{\text{PT}\to\text{IR}} y \,\middle|\, \begin{matrix} - \\ 0, 0 \end{matrix}\right) G_{0,1}^{1,0}\left(\alpha\lambda_{\text{PT},\text{IR}} y x \,\middle|\, \begin{matrix} - \\ 0 \end{matrix}\right) dy \\
&= 1 - \frac{\lambda_{\text{PT}\to\text{IR}}}{\alpha\lambda_{\text{PT},\text{IR}} x} \exp\left(-\frac{\lambda_{\text{PT},\text{IR}} x}{\Psi}\right) \\
\mathcal{H}_{1,0;0,2;1,1}^{0,1;2,0;0,1}&\left(\begin{matrix} 0 : 1, 1 \\ - \end{matrix} \,\middle|\, \begin{matrix} - \\ (0,1);(0,1) \end{matrix} \,\middle|\, \begin{matrix} 1, 1 \\ 0, 1 \end{matrix} \,\middle|\, \frac{\lambda_{\text{PT}\to\text{IR}}}{\alpha\lambda_{\text{PT},\text{IR}} x}; \frac{\Psi}{\lambda_{\text{PT},\text{IR}} x^2}\right),
\end{aligned}
\tag{25}
$$

where the last step can be attained based on two identities [50, eq 2.3] and [51, eq. 1.7.1] and $\mathcal{H}_{:::}^{:::}\left(\begin{smallmatrix}:::\\:::\end{smallmatrix}\middle|\begin{smallmatrix}:::\\:::\end{smallmatrix}\middle|\begin{smallmatrix}:::\\:::\end{smallmatrix}\middle|\cdot;\cdot\right)$ is the bi-variate Fox-H function.

**3.2.2 Average BLER analysis.** Having obtained the CDF of $\gamma_{\text{IR}}$ in hand, we are next so excited to derive the average BLER of IR by making use of the relation in (11).

**Theorem 2**. *A closed-form solution for the average BLER of the IoT sensor receiver can be formulated as*

$$\epsilon_{\mathrm{IR}} \simeq \sum_{k=1}^{K} \frac{\pi}{2K} \sqrt{1 - \psi_k^2} F_{\gamma_{\mathrm{IR}}} \left( \frac{\psi_k}{2\zeta_{\mathrm{IR}}} + \kappa_{\mathrm{IR}} \right), \tag{26}$$

*where K represents the complexity-accuracy trade-off parameter while* $\psi_l = \cos(\pi(2k - 1)/[2K])$.

*Proof.* By injecting the developed CDF in (25) into the expression in (11), we can establish the average BLER formulation as

$$
\epsilon_{\mathrm{IR}} \simeq 1 \quad -\zeta_{\mathrm{IR}} \frac{\lambda_{\mathrm{PT \to IR}}}{\alpha \lambda_{\mathrm{PT,IR}}} \int_{\upsilon_{\mathrm{IR}}}^{\vartheta_{\mathrm{IR}}} \frac{1}{x} \exp\left(-\frac{\lambda_{\mathrm{PT,IR}} x}{\Psi}\right)
$$
$$
\times \mathcal{H}_{1,0;0,2;1,1}^{0,1;2,0;0,1} \left( \begin{array}{c} 0:1,1 \\ - \end{array} \bigg| \begin{array}{c} - \\ (0,1);(0,1) \end{array} \bigg| \begin{array}{c} 1,1 \\ 0,1 \end{array} \bigg| \frac{\lambda_{\mathrm{PT \to IR}}}{\alpha \lambda_{\mathrm{PT,IR}} x}; \frac{\Psi}{\alpha \lambda_{\mathrm{PT,IR}} x^2} \right) dx. \tag{27}
$$

However, due to the intractability of the H-fox function, we tackle the integral in (27) by applying the Gaussian-Chebyshev quadrature method.

Having achieved Theorem 2, we are interested in concluding that a unique function of all system parameters can characterize the average BLER of IR analytically. Nevertheless, the involvement of the H-fox form in $F_{\gamma_{\mathrm{IR}}}(\cdot)$ in (25) makes the analysis quite complex; thus, it raises the question of figuring out a simpler BLER formulation. In response to this question, we make use of the equivalent infinitesimal $1/x \simeq 0$ as $x \to \infty$ to simplify the CDF of $\gamma_{\mathrm{IR}}$ in (21) as

$$
F_{\gamma_{\mathrm{IR}}}(x) \overset{\Psi \to \infty}{=} 1 - 2\lambda_{\mathrm{PT \to IR}} \exp\left(-\frac{\lambda_{\mathrm{PT,IR}} x}{\Psi}\right) \int_0^\infty \exp(-\alpha \lambda_{\mathrm{PT,IR}} yx) \mathcal{K}_0\left(2\sqrt{\lambda_{\mathrm{PT \to IR}} y}\right) dy
$$
$$
= 1 - \exp\left(\frac{-\lambda_{\mathrm{PT,IR}} x}{\Psi}\right) G_{1,2}^{2,1} \left( \frac{\lambda_{\mathrm{PT \to IR}}}{\lambda_{\mathrm{PT,IR}} x\alpha} \bigg| \begin{array}{c} - \\ 1,1 \end{array} \right), \tag{28}
$$

where the last step is derived using the Meijer-G function transformations in (23) and (24) before applying the standard form in [48, eqs. (7.811.1) and (9.31.2)]. It is clear that the result in (28) has a simpler format compared to that of (25). However, the complex nature of the Meijer-G function still limits deriving the average BLER. To end this, we rely on the first-order Riemann integral approximation [11, Eq. (15)], which leads to

$$
\tilde{\epsilon}_{\mathrm{IR}} \overset{\Psi \to \infty}{\simeq} \zeta_{\mathrm{IR}} \int_{\upsilon_{\mathrm{IR}}}^{\vartheta_{\mathrm{IR}}} F_{\gamma_{\mathrm{IR}}}(x) dx = \zeta_{\mathrm{IR}} (\vartheta_{\mathrm{IR}} - \upsilon_{\mathrm{IR}}) F_{\gamma_{\mathrm{IR}}} \left( \frac{\vartheta_{\mathrm{IR}} + \upsilon_{\mathrm{IR}}}{2} \right)
$$
$$
= 1 - \exp\left(\frac{-\lambda_{\mathrm{PT,IR}} \kappa_{\gamma_{\mathrm{IR}}}}{\Psi}\right) G_{1,2}^{2,1} \left( \frac{\lambda_{\mathrm{PT \to IR}}/\alpha}{\lambda_{\mathrm{PT,IR}} \kappa_{\gamma_{\mathrm{IR}}}} \bigg| \begin{array}{c} - \\ 1,1 \end{array} \right). \tag{29}
$$

From the above formulation, it is interesting to show that the BLER performance exponentially decreases with the $\Psi$. Especially, when we let $1/\Psi = 0$, an interesting result can be deduced from (29) that the average BLER will converge to a saturation, which is determined

by

$$\tilde{\epsilon}_{\text{IR}} \overset{\Psi \to \infty}{\simeq} 1 - G_{1,2}^{2,1}\left(\frac{\lambda_{\text{PT} \to \text{IR}}/\alpha}{\lambda_{\text{PT,IR}}\kappa_{\gamma_{\text{IR}}}} \,\middle|\, \begin{matrix} - \\ 1, 1 \end{matrix}\right). \tag{30}$$

With this formulation, it is interesting to show that increasing $N_{\text{IR}}$ scales down the Meijer-G function, which is equivalent to increasing the BLER. In contrast, increasing $L_{\text{IR}}$ scales up the Meijer-G function, which then decreases the BLER.

## 4 Numerical results and discussions

This section provides some illustrative numerical results using Monte-Carlo simulations to validate our developed mathematical framework, where the number of used channel realization samples is $10^5$. Without loss of generality, we consider the specific parameters for Rayleigh channels as follows: $\lambda_{\text{PT,PR}} = 2$, $\lambda_{\text{PT,BD}} = 4$, $\lambda_{\text{PT,IR}} = 3$, and $\lambda_{\text{BD,IR}} = 5$ (channel modeling has been early described at Eq 1). Unless otherwise specified, the key simulation parameters related to packet designs and transmit SNR are set as follows: $\alpha = 0.5$, $L_{\text{RX}} = 256$ c.u, $N_{\text{PR}} = 300$ bits, $N_{\text{IR}} = 80$ bits, and $\Psi = 25$ dB.

Fig 2 shows the average BLER versus $\alpha$. We look at the case where PT sends 300 bits of data to PR, while BD produces 80 bits of data for command control sync. It is observed that the error performance of PR remains constant with respect to $\alpha$ since its receiving signal does not gain any backscattering signal from BD. Meanwhile, IR's error performance tends to reduce with a small value of $\alpha$ and then become saturated. This is because on the one hand, scaling up

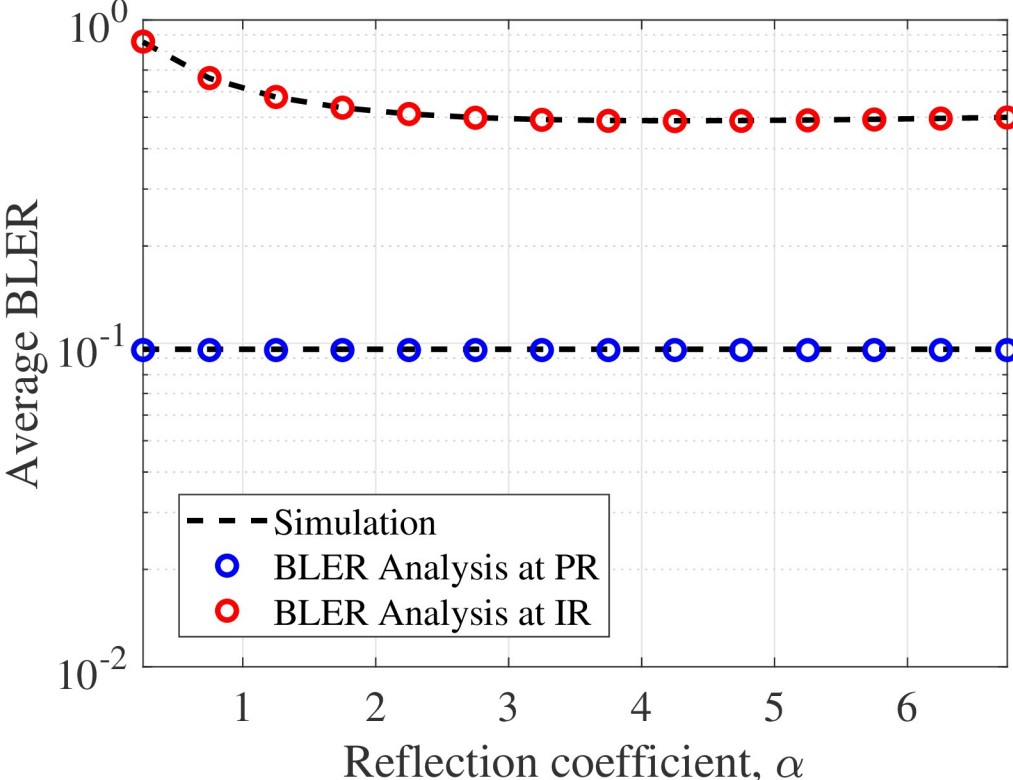

**Fig 2. Impact of $\alpha$ on BLER.**

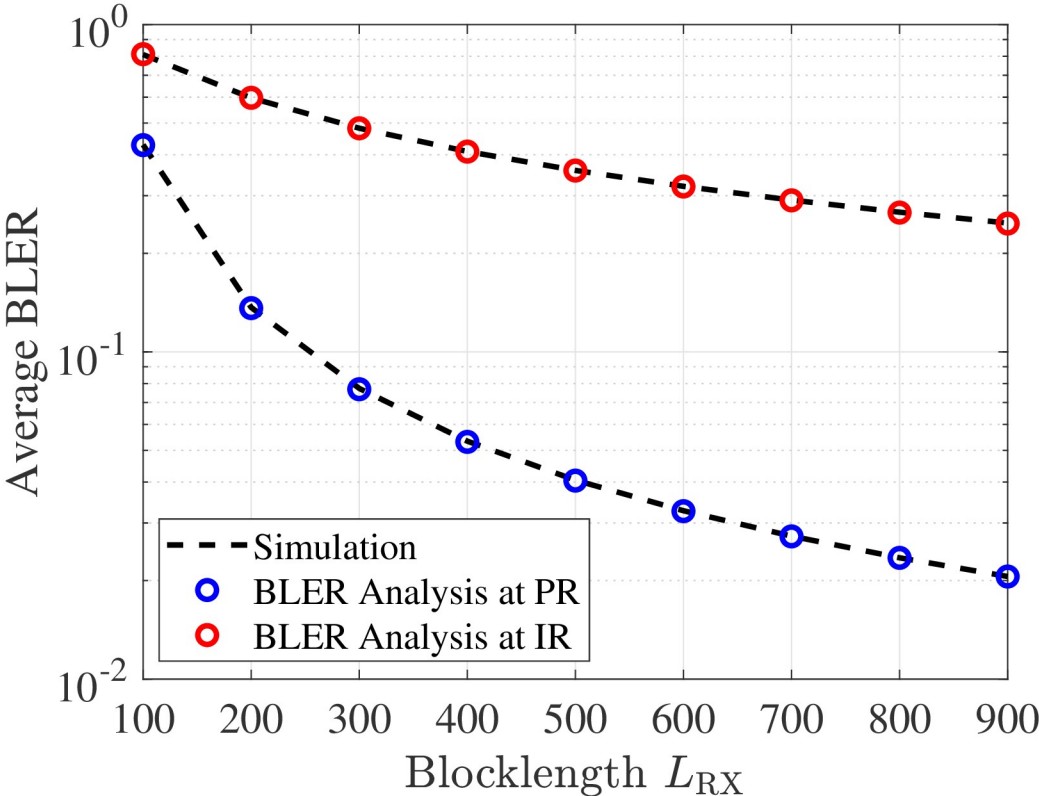

**Fig 3. Impact of $L_{RX}$ on BLER.**

$\alpha$ improves the received SNR signal to decode $c(t)$ in (6) but decreases the received SINR signal to decode $x(t)$ in (5) on the other hand. Recall that the SIC procedure dominates the decoding process at IR. Taking these together therefore explains why increasing $\alpha$ does not yield any error performance improvement.

Fig 3 explores the impact of block-length $L_{RX}$ on the average BLER. From the figure, we can see that while the error performance of PR reduces considerably with an increase in $L_{RX}$, that of IR decreases relatively low. The reasons are interpreted as follows: 1) for PR, the received SNR given in (3) does not suffer from any SIC, which gives PR a chance to decode $x(t)$ without interference. Thus, it is safe to conclude that the more block-length (channel use) of the information transmission, the higher the reliability of the communication channel. Recall that, such phenomenon completely accords the analysis for the developed expression in (16). 2) for IR, its decoding process takes place in two phases of decoding $x(t)$ and $c(t)$, respectively. Thus, this process will increase an expected error during decoding $c(t)$, making the error performance of IR to be reduced slowly. Recall that such a phenomenon can be directly explained from (29), where increasing $L_{PR}$ reduces the exponential component but scales up the Meijer-G component accordingly.

Fig 4 showcases the average BLER against data amount sent by PT and BD. As observed, conveying more data over a fixed channel use to the receiving node causes more errors during communication, thereby leading to an increase in the BLER trend. These trends are perfectly matched with our analyses for the expression in (16) and (29). In this case, a more channel should be allocated to boost reliable communication. Yet, this assignment might not be

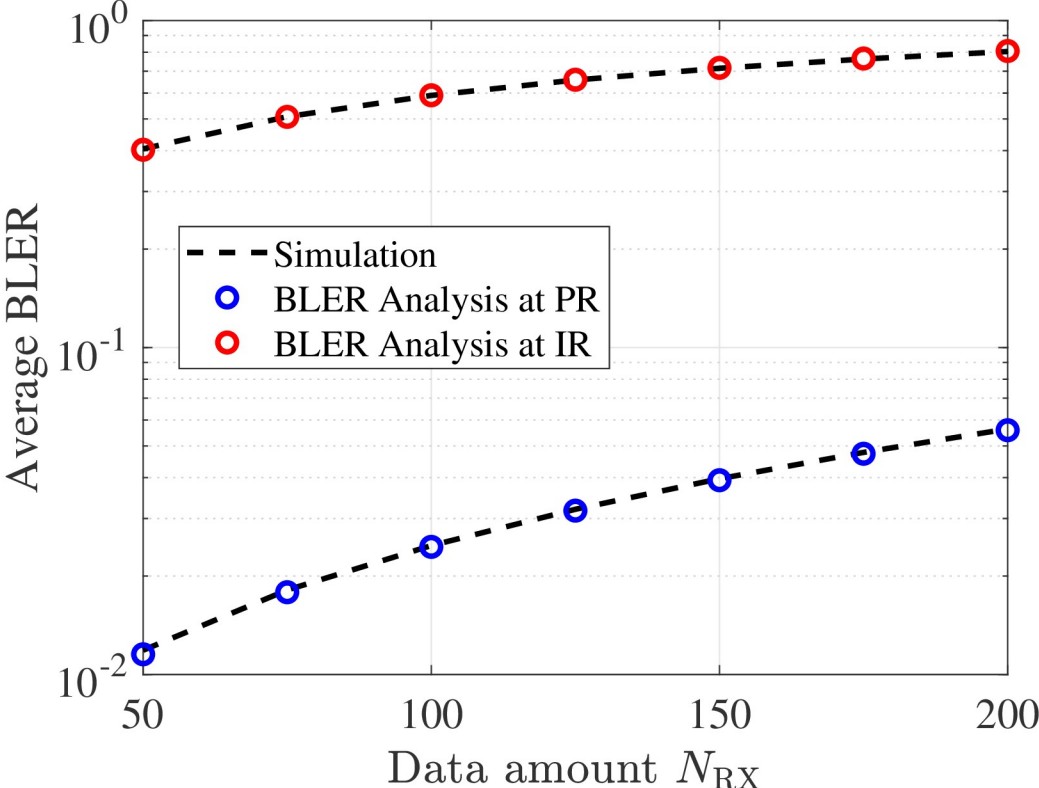

**Fig 4. Impact of $N_{RX}$ on BLER, with $L_{RX}$ = 100 c.u.**

beneficial to the systems as it is equivalent to an increase in the transmission latency. Therefore, it is necessary to consider balancing such configuration for each application critically.

Fig 5 depicts the impact of Ψ on the average BLER. Overall, we can find that increasing Ψ significantly improves the average BLER of PR and the trend is linearly decreasing. Clearly, this observation perfectly matches up with the developed formula in (16), where the error performance also becomes zero when we take into consideration 1/Ψ = 0. Meanwhile, varying PR only improves the BLER of IR in moderate SNR but saturates at high SNR, which perfectly agrees with the conclusion drawn on (30). On this basis, in order to improve the BLER of IR, we should take care of both increasing Ψ in conjunction with an increasing number of block-lengths.

## 5 Conclusion

In this work, we have studied the performance of symbiotic backscatter communication systems with short-packet transmissions. Particularly, aiming to characterize the system performance without performing any simulation, we derived closed-form approximate and asymptotic expressions of the average BLER for both the primary receiver and IoT sensor. These mathematical frameworks enable us to directly assess the system performance by the key parameters of the transmit power, fading parameters, data amount, and packet length. To ensure the correctness of the developed mathematical framework, we produced some illustrative simulation results based on the Monte-Carlo simulation while comparing the actual impact of system parameters on the BLER behaviour over the analysis outcome.

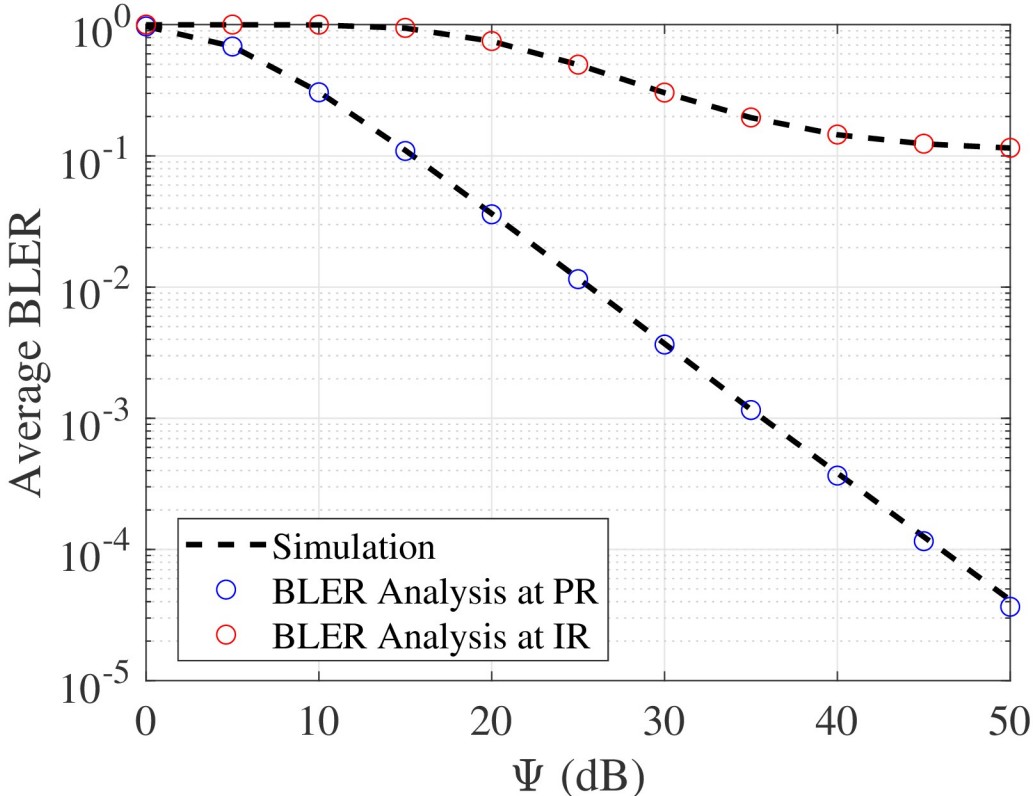

**Fig 5. Impact of Ψ on BLER, with $L_{\mathrm{RX}}$ = 100 c.u and $N_{\mathrm{RX}}$ = 150 bits.**

## Supporting information

**S1 File.**
(ZIP)

## Author Contributions

**Conceptualization:** Quang Vinh Do.

**Formal analysis:** Bui Vu Minh, Quang-Sang Nguyen.

**Methodology:** Quang-Sang Nguyen.

**Validation:** Byung-seo Kim.

**Writing – original draft:** Quang Vinh Do, Bui Vu Minh.

**Writing – review & editing:** Byung-seo Kim.

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
