## [Decision Letter · Decision Letter 0]

14 May 2024

PONE-D-24-11836Analysis of Symbiotic Backscatter Empowered Wireless Sensors Network with Short-Packet CommunicationsPLOS ONE

Dear Dr. Do,

Thank you for submitting your manuscript to PLOS ONE. After careful consideration, we feel that it has merit but does not fully meet PLOS ONE’s publication criteria as it currently stands. Therefore, we invite you to submit a revised version of the manuscript that addresses the points raised during the review process.

We look forward to receiving your revised manuscript.

Kind regards,

Kamarul Ariffin Noordin, Ph.D.

Academic Editor

PLOS ONE

“This work was supported by the National Research Foundation of Korea (NRF) grant funded by the Korea government (MSIT) (No. 2022R1A2C1003549).”

5. We note that Figure 1 in your submission contain copyrighted images. All PLOS content is published under the Creative Commons Attribution License (CC BY 4.0), which means that the manuscript, images, and Supporting Information files will be freely available online, and any third party is permitted to access, download, copy, distribute, and use these materials in any way, even commercially, with proper attribution. For more information, see our copyright guidelines: http://journals.plos.org/plosone/s/licenses-and-copyright.

Reviewers' comments:

Reviewer's Responses to Questions

**Comments to the Author**

1. Is the manuscript technically sound, and do the data support the conclusions?

Reviewer #1: Yes

2. Has the statistical analysis been performed appropriately and rigorously? 

Reviewer #1: Yes

3. Have the authors made all data underlying the findings in their manuscript fully available?

Reviewer #1: Yes

4. Is the manuscript presented in an intelligible fashion and written in standard English?

Reviewer #1: Yes

5. Review Comments to the Author

Reviewer #1: Authors investigated the feasibility of Backcom and SPC for symbiotic wireless sensor networks by analyzing the system performance. Specifically, we provide a highly approximated mathematical framework for evaluating the block-error rate (BLER) performance, followed by some useful asymptotic results.

Paper is well written and idea is clear, however i have some major comments.

1: Eq 16 need proof.

2:Simulation and channel modelling need more detail.

3: Some important references are missing.

a:X. Li et al., "Physical Layer Security for Wireless-Powered Ambient Backscatter Cooperative Communication Networks," in IEEE Transactions on Cognitive Communications and Networking, vol. 9, no. 4, pp. 927-939, Aug. 2023

b:https://www.techscience.com/CMES/v138n1/54237

6. PLOS authors have the option to publish the peer review history of their article (what does this mean?). If published, this will include your full peer review and any attached files.

Reviewer #1: No

---

## [Author Response · Author response to Decision Letter 0]

30 May 2024

Please kindly check the attached Response to Reviewers letter.

---

## [Decision Letter · Decision Letter 1]

4 Jul 2024

Analysis of Symbiotic Backscatter Empowered Wireless Sensors Network with Short-Packet Communications

PONE-D-24-11836R1

Dear Dr. Do,

We’re pleased to inform you that your manuscript has been judged scientifically suitable for publication and will be formally accepted for publication once it meets all outstanding technical requirements.

Kind regards,

Kamarul Ariffin Noordin, Ph.D.

Academic Editor

PLOS ONE

Additional Editor Comments (optional):

Reviewers' comments:

Reviewer's Responses to Questions

**Comments to the Author**

1. If the authors have adequately addressed your comments raised in a previous round of review and you feel that this manuscript is now acceptable for publication, you may indicate that here to bypass the “Comments to the Author” section, enter your conflict of interest statement in the “Confidential to Editor” section, and submit your "Accept" recommendation.

Reviewer #1: (No Response)

2. Is the manuscript technically sound, and do the data support the conclusions?

Reviewer #1: (No Response)

3. Has the statistical analysis been performed appropriately and rigorously? 

Reviewer #1: (No Response)

4. Have the authors made all data underlying the findings in their manuscript fully available?

Reviewer #1: (No Response)

5. Is the manuscript presented in an intelligible fashion and written in standard English?

Reviewer #1: (No Response)

6. Review Comments to the Author

Reviewer #1: Authors have addressed most of comments

7. PLOS authors have the option to publish the peer review history of their article (what does this mean?). If published, this will include your full peer review and any attached files.

Reviewer #1: No

---

## [Editor Report · Acceptance letter]

10 Jul 2024

PONE-D-24-11836R1 

PLOS ONE

Dear Dr. Do, 

I'm pleased to inform you that your manuscript has been deemed suitable for publication in PLOS ONE. Congratulations! Your manuscript is now being handed over to our production team.

Kind regards, 

on behalf of

Dr. Kamarul Ariffin Noordin 

Academic Editor

PLOS ONE